# Electrical conductivity and magnetic dynamos in magma oceans of Super-Earths

François Soubiran [1,2] & Burkhard Militzer[1,3]

Super-Earths are extremely common among the numerous exoplanets that have been discovered. The high pressures and temperatures in their interiors are likely to lead to long-lived magma oceans. If their electrical conductivity is sufficiently high, the mantles of Super-Earth would generate their own magnetic fields. With ab initio simulations, we show that upon melting, the behavior of typical mantle silicates changes from semi-conducting to semi-metallic. The electrical conductivity increases and the optical properties are substantially modified. Melting could thus be detected with high-precision reflectivity measurements during the short time scales of shock experiments. We estimate the electrical conductivity of mantle silicates to be of the order of $100\,\Omega^{-1}\,cm^{-1}$, which implies that a magnetic dynamo process would develop in the magma oceans of Super-Earths if their convective velocities have typical values of 1 mm/s or higher. We predict exoplanets with rotation periods longer than 2 days to have multipolar magnetic fields.

[1] Department of Earth and Planetary Science, University of California, Berkeley, CA 94720, USA. [2] École Normale Supérieure de Lyon, Université Lyon 1, Laboratoire de Géologie de Lyon, CNRS UMR5276, Lyon Cedex 07 69364, France. [3] Department of Astronomy, University of California, Berkeley, CA 94720, USA. Correspondence and requests for materials should be addressed to F.S. (email: francois.soubiran@ens-lyon.org)

Space missions dedicated to the search for exoplanets such as CoRoT or Kepler led to the discovery of a large set of planets outside our Solar system. These exoplanets are diverse in terms of radius, mass, and distance to the host-star[1], which makes it challenging to identify a unique planetary formation mechanism[2]. Among these numerous planets, a significant proportion are smaller than three Earth radii[3]. Their mean density implies that they are to large degree made of silicates[4,5], although not all are entirely rocky[6]. Despite the development of numerous Super-Earth models[7,8], their interior structure remains uncertain but they are likely to be significantly hotter than the Earth and to reach much higher pressures[9,10]. One thus expects them to have sizable magma oceans that persist over long period of time. The properties of magma oceans are poorly constrained but one expects them to be rapidly convecting. If they would also be good electrical conductors, they would generate a dynamo and produce a magnetic field[11], which is relevant for the habitability of exoplanets[12].

To answer the question whether magma oceans can contribute to the magnetic field generation of exoplanets, one needs to determine the phase diagram, equations of state (EOS) and transport properties of typical mantle silicates such as MgO, SiO₂, or MgSiO₃. These materials have been studied using ab initio simulations, which identified various high pressure solid phases and a melting temperature of 10,000–12,000 K in the megabar regime for MgO but only approximately 5000–8000 K for MgSiO₃[13–21]. Experimentally, static high pressure experiments[22,23] and ramp shock compressions[24] confirmed the existence of some of the predicted solid phases. The melting line has been studied with single and decaying shocks. While some discrepancies among the melting line measurements and with the numerical predictions remain to be resolved[25–34], it is very likely that Super-Earths would be sufficiently hot to have thick magma oceans in their interiors[35].

While silicates are insulators and optically transparent at ambient conditions[36,37], shock experiments observed a non-zero reflectivity in the liquid phase[26,28,32–34]. Ab initio simulations have also reported an increase of reflectivity and conductivity in the liquid silicate at high temperature[15,19]. It has not yet been studied how electronic structure and transport properties change when silicates melt at high pressure.

In this article we characterize the electronic properties of the solid and liquid silicates with ab initio simulations for the same pressure and temperature. We compare our results with shock experiments and propose that reflectivity measurements as an indirect approach to detect melting on the short timescale of shock experiments. Finally we explore how the electronic properties of liquid silicate affect the dynamo processes in Super-Earth and characterize the conditions for magma oceans to generate their own magnetic fields.

## Results

**Electronic gap closure upon melting.** Using ab initio simulations, we compared the properties of the solid and the liquid phases of MgO, SiO₂, and MgSiO₃. We performed simulations for the liquid and the solid under conditions that are close to the solid–liquid phase boundary and to the Hugoniot pressure–temperature path. For MgO, we choose 12,000 K and 470 GPa based on estimates of the B2 melting[27] and the principal Hugoniot[26]. For SiO₂ we used the experimental results on decaying shocks in stishovite[33] and chose 500 GPa and 9000 K, the stable solid phase being the pyrite[16,17,22]. For MgSiO₃ we chose 240 GPa and 7000 K, which corresponds to the intersection of the predicted melting curve[20] and the Hugoniot started in the single crystal[34]. Under these conditions, the solid stable phase is expected to be post-perovskite[38]. For the three materials, we performed both liquid and solid simulations under the same conditions in order to determine the effect of the phase change on the electronic properties. Figure 1 shows the P–T conditions considered and how they compare with the phase diagrams and the Hugoniot curves.

In Fig. 2 we displayed the electronic density of states (DOS) centered on the Fermi level. For the three materials, we observe the existence of an energy gap of the order of 2–4 eV. These values are significantly lower than those at ambient conditions. For instance, the band gap is 7.8 eV in rock salt MgO, 10.4 eV in quartz[39], and 8.9 eV in orthoenstatite MgSiO₃ (ref. [40]). But the compression of the material induces hybridizations of the highest orbitals and a continuum lowering which reduces the size of the gap[41].

According to our calculations, the eigenstates are nearly fully occupied in the valence band but one can see a small fraction of partially occupied states in the conduction band. This is the typical behavior for a semi-conducting material and we can expect to observe a low direct current (DC) conductivity and a higher value above a frequency equivalent to the gap energy. Under the same P–T conditions, we clearly observe in Fig. 2 that, for the three materials, the gap is closed in the liquid phase. There

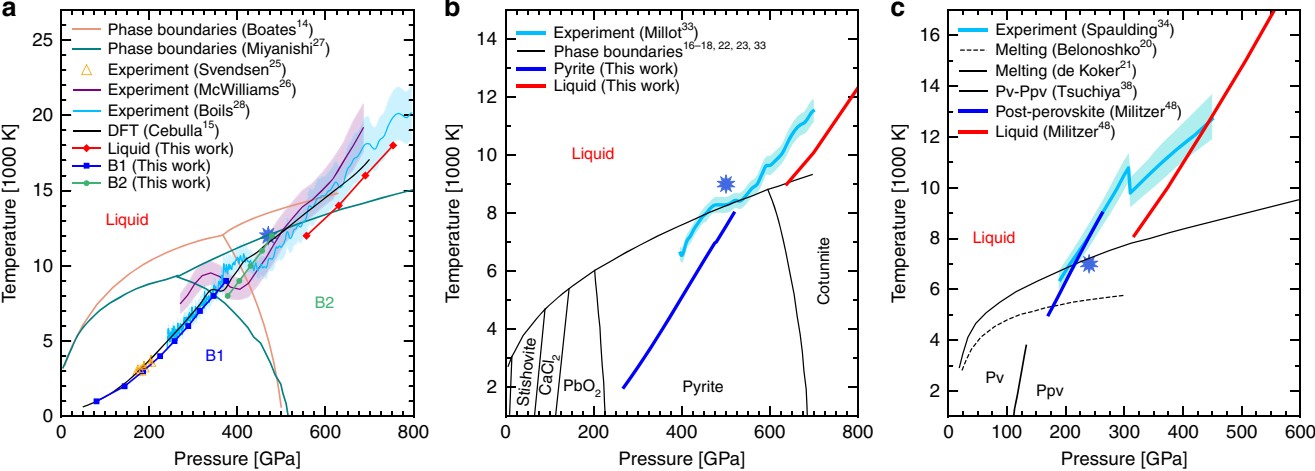

**Fig. 1** Hugoniot curves and phase diagram of silicates. Phase diagram of **a** MgO[14,27], **b** SiO₂[16–18,22,23,33], and **c** MgSiO₃[20,21,38] are plotted. We added our predictions for the Hugoniot curves in the different phases as well as other first-principles predictions[15,48] and experimental shock results[25,26,33,34]. The shaded areas represent the experimental uncertainty. The blue stars indicate the P–T conditions we chose for the liquid/solid comparison

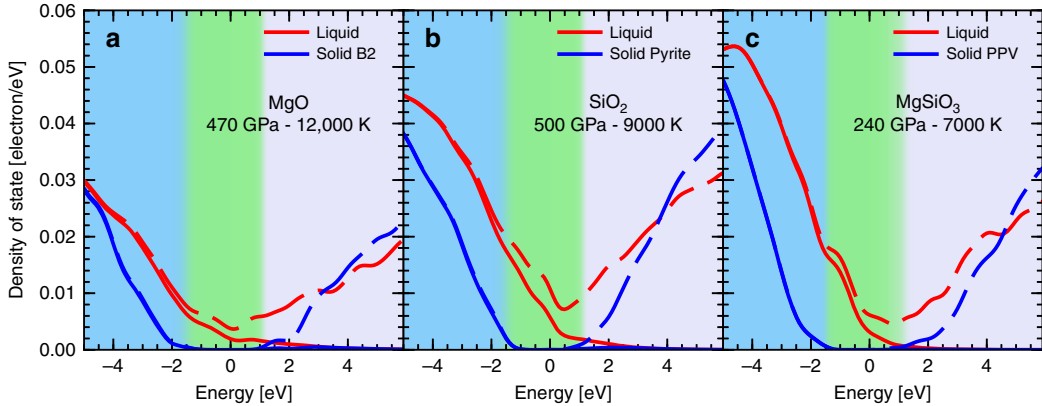

**Fig. 2** Electronic density of states (DOS) of silicates. The liquid (red) and the solid (blue) DOSs with the Fermi level shifted to 0 eV are plotted for **a** MgO, **b** SiO$_2$, and **c** MgSiO$_3$. The dash-dotted lines are the complete DOSs and the full the occupied DOSs only. The conditions were chosen so that they are close to the respective Hugoniot curve[26,33,34] and the expected phase boundary[20,27,33] as shown by the blue stars in Fig. 1. The blue shaded region represents the valence band in the solid, the green one is the gap, and the light purple one is the conduction band

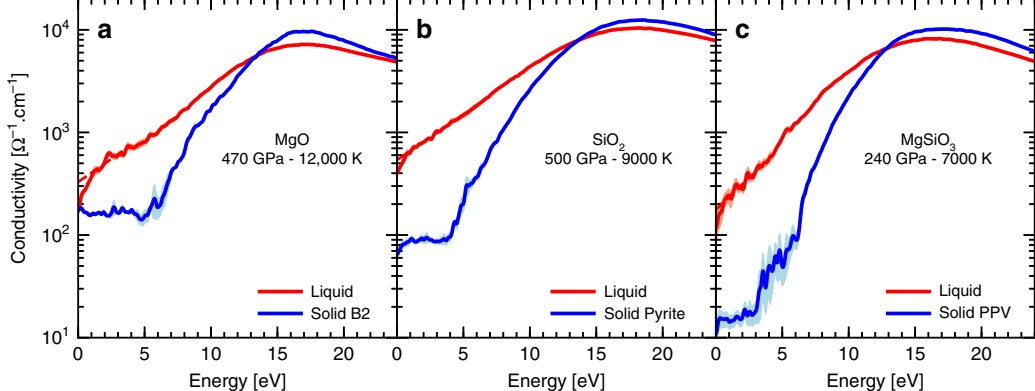

**Fig. 3** Electric conductivity. The liquid (red) and the solid (blue) electric conductivities as a function of the excitation energy are plotted for **a** MgO, **b** SiO$_2$, and **c** MgSiO$_3$. The conditions are the same as those of Fig. 2. The shaded region is an indication of the one-sigma statistical uncertainty. We plotted in dashed lines the extrapolation of the conductivity in the liquid phase towards the direct current (DC) value (see the Methods section)

is a continuum of eigenstates below and above the Fermi level suggesting a conducting behavior. However, because the number of eigenstates near the Fermi level is relatively small we expect the liquid to be poorly conducting and to behave as a semimetal. Nevertheless, because the gap is closed we predict the conductivity at low frequency to be much higher in the liquid than in the solid.

**Electronic transport properties upon melting**. We further determined the electric conductivity in the liquid and solid phases. The comparison is shown in Fig. 3. As with the DOS analysis, for the three solid materials, we observe a clear semi-conducting behavior with a low and frequency-independent conductivity up to an excitation energy that corresponds to the energy gap in the DOS. The low-frequency plateau has a conductivity of $160\,\Omega^{-1}\,cm^{-1}$ in B2 MgO, $90\,\Omega^{-1}\,cm^{-1}$ in pyrite SiO$_2$, and $15\,\Omega^{-1}\,cm^{-1}$ in post-perovskite MgSiO$_3$.

In the liquid, we observe a slightly higher conductivity than the solid phases at low frequency and no plateau. The conductivity increases with excitation energy up to a maximum that is close to the values in the solid phase. This frequency dependency is the typical behavior of a semimetal[42] as anticipated from the DOS analysis. We found that under the conditions presented in Fig. 3, the DC conductivity is $336\,\Omega^{-1}\,cm^{-1}$ for liquid MgO, $488\,\Omega^{-1}\,cm^{-1}$ for liquid SiO$_2$—which is very comparable to values computed by Scipioni et al.[43]—and $216\,\Omega^{-1}\,cm^{-1}$ for liquid

MgSiO$_3$. In each case, the conductivity is significantly larger in the liquid than in the solid with a difference of a factor of 2 up to 14.

We applied the Kubo–Greenwood formula to determine the thermal conductivity. We do not attempt to compute the ionic contributions, because we anticipate the electronic contribution to be dominant due to the significant electric conductivity in the liquid. Following a very similar procedure as for the electrical conductivity, we found 31 W/m/K for MgO, 28 W/m/K for SiO$_2$, and 8 W/m/K for MgSiO$_3$ under the conditions mentioned in Fig. 3, in the liquid phase. For comparison, at the core-mantle boundary (135 GPa and 3700 K (ref. [44])) the thermal conductivity of post-perovskite is estimated to be 16.8 W/m/K (ref. [45]). Under the same conditions, the thermal conductivity of iron has been measured between 33 W/m/K (ref. [46]) and 226 W/m/K (ref. [47]). The discrepancy in the thermal conductivity measurements underlines the difficulty of such experiments and the still large uncertainty on the iron conductivity in Earth' core. It is interesting to note that by combining the electrical and thermal conductivities we find that the Lorenz number for liquid silicates is significantly higher than the theoretical value of $2.44 \times 10^{-8}\,W\Omega/K^2$ (ref. [42]). We obtain $7.7 \times 10^{-8}\,W\Omega/K^2$ for MgO, $6.4 \times 10^{-8}\,W\Omega/K^2$ for SiO$_2$, and $5.3 \times 10^{-8}\,W\Omega/K^2$ for MgSiO$_3$. These differences suggest that using the Wiedemann–Franz law to determine the thermal conductivity based on an experimentally or theoretically derived electrical conductivity might lead to a significant underestimation

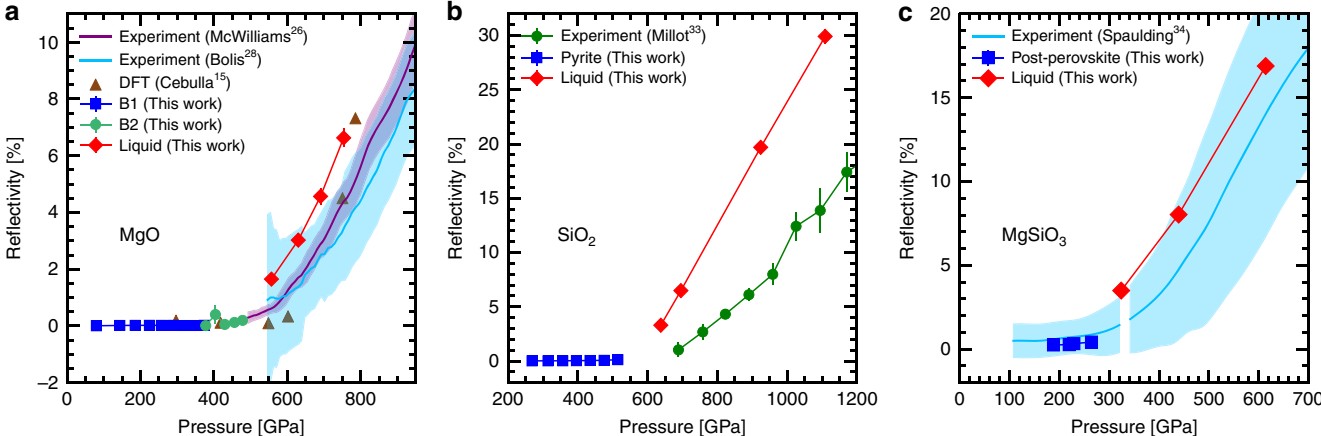

**Fig. 4** 532 nm reflectivity along the principal Hugoniot. We plotted our calculated reflectivity (with a one-$\sigma$ error bar) in the different phases of **a** MgO, **b** SiO$_2$, and **c** MgSiO$_3$ along their respective Hugoniot curves as described in Fig. 1. We added previous numerical results for MgO[15], and systematically compared with available experimental results[26,28,33,34] including their uncertainty

of the thermal conductivity in liquid silicates. This is however not a surprising result since the silicates behave as semimetals and not as highly conducting metals for which the Wiedmann–Franz law is verified.

**Reflectivity along the Hugoniot.** In order to compare our findings to recent high-pressure experiments, we computed the Hugoniot curve for MgO, started from a single crystal of B1-MgO[26,28], for SiO$_2$ started from the stishovite phase[33], and for MgSiO$_3$ started from a single crystal of enstatite[28,34]. We used the phase diagrams available in the literature[21,27,33] to choose the relevant phase for the Hugoniot calculations as can be seen in Fig. 1. For MgO, the computed Hugoniot identifies three distinct branches in the $P-T$ plane for the three different phases while experiments only identify two. The calculated Hugoniot curve in the B1 phase agrees very well with two sets of experiments while McWilliams et al.[26] measured it 1000 K hotter. For SiO$_2$, the evolution of the Hugoniot curve as pressure increases is the same in the experiment and in the calculated curves but the latters are systematically shifted by about 100 GPa towards higher pressure. The reason of such a shift is unclear. In the case of MgSiO$_3$, the post-perovskite calculated Hugoniot curve from ref. [48] matches the low pressure branch of Spaulding et al.'s experiment. The experimental Hugoniot on the second branch has a different evolution than the predicted behavior for the liquid but they seem to retrieve similar values at higher temperatures.

Along the computed Hugoniot curves, we explored the reflectivity at 532 nm as plotted in Fig. 4. The computed reflectivity in B1 and B2 MgO is 0 or almost 0. The reflectivity is much more significant in the liquid phase and we observe a similar evolution as a function of the pressure as in the two shock experiments. The computed values are however about 1 percentage point above the experimental values. We do not observe a reflectivity as sharply evolving with the pressure as the one predicted by Cebulla et al.[15]. The observations are very similar in SiO$_2$ with a zero reflectivity in the pyrite phase and a higher reflectivity in the liquid. We have however a significant deviation from the experimental value, which could be related to the fact that our predicted Hugoniot is shifted towards higher pressures than observed in the experiment. The MgSiO$_3$ Hugoniot curve also exhibits a non-reflecting solid phase but an increasing reflectivity in the liquid phase as the pressure increases. The agreement with the experiment is very good.

It is interesting to observe that the three silicates we studied exhibit the same feature with non-reflecting solid phases but reflecting liquid phases. It could be an indication of the phase in which we are during an experiment: if the reflectivity is measurable then we are likely in the liquid phase while if the reflectivity is 0, it is the solid phase. However, the computed Hugoniot for the three materials show a discontinuity in pressure that should not exist in the experiment. In fact, our simulations assumed a single phase and resulted in a wide pressure gap between the solid Hugoniot curve and the liquid one. In reality, there must be a range of pressures where the solid and the liquid are in equilibrium. In that case there could be a continuum between a zero reflectivity and a measurable non-zero value. One would have to look for changes in the slope of the reflectivity curve to identify the phase transition.

## Discussion

Because large and/or young Super-Earths are potentially relatively hot, they could have a large magma ocean in their envelope. It is interesting to note that the computed values of the thermal conductivity for the three types of liquid silicates we explored are not very high, which means that large-scale convection is likely to develop[49] in a magma ocean. At the same time, the electrical conductivity is not negligible and might allow for a small enough magnetic diffusivity, and thus for a dynamo process to occur within the magma ocean. With values of the electrical conductivity between 200 and 500 $\Omega^{-1}$ cm$^{-1}$ we find the magnetic diffusivity to be $\eta = 1/(\mu_0 \sigma) \sim 16 - 40$ m$^2$/s. Based on dynamo simulations, it is known that a dynamo process can develop in a large convective zone if the magnetic Reynolds number $R_m = \frac{vl}{\eta}$ is above a critical value of about 40 (ref. [11]). With the aforementioned values of the magnetic diffusivity and a typical eddy size of $l \sim 10^6$ m, we find a minimal convective velocity of the order of $v \sim 0.6-1.6$ mm/s. This is a realistic value for a convective magma ocean, especially when compared to a typical value of a few tenths of mm/s in the Earth's outer core[50]. In other words, if a Super-Earth has a magma ocean of a thickness of 1000 km, with a typical convective velocity of at least 1 mm/s, it would produce a significant dynamo process.

The dynamo simulations also provided scaling laws that allow to determine the structure of the magnetic field. For instance, the magnetic fields on Earth is dipolar but Neptune and Uranus have a multipolar magnetic field. To determine if the magnetic field

generated by the magma ocean is dipolar, one needs to estimate the local Rossby number[51] which scales as

$$\text{Ro}_l = 0.11 \left( \frac{\text{Ro}}{1.23} \right)^{1.22} E^{-1/3} \text{Pr}^{1/5} P_m^{-1/5}, \qquad (1)$$

with Ro, $E$, Pr, and $P_m$, respectively, the Rossby, Ekman, Prandtl. and magnetic Prandtl numbers as defined in ref. [51]. To estimate these different dimensionless numbers, we chose a typical density of $\rho \sim 6$ g/cm$^3$, a kinematic viscosity of $\nu \sim 10^{-5}$ m$^2$/s, and a heat capacity at constant pressure of $C_P \sim 800$ J/kg/K[52]. These values are merely order of magnitudes and a complete characterization of the silicated materials under high pressures is needed for a more accurate determination of these quantities. The velocity was assumed to be of the order of 1 mm/s, corresponding to the lower limit for the dynamo to develop. For the typical angular velocity, we considered planets that would be in the habitable zone of M-dwarfs which are favorite candidates for the detection of Super Earth. Since they are very close to there stars, these planets are very likely to be tidally locked and thus a typical period of rotation and revolution is of the order of 10 days[53] leading to an angular velocity of $\Omega \sim 7 \times 10^{-6}$ s$^{-1}$. All combined, we obtain Ro$_l$ $\sim 0.4$ which is above the maximum value of 0.1 for which dynamo simulations predict a dipole dominated magnetic field structure[11]. It is interesting to note that the local Rossby number is proportional to the convective velocity. This means that a higher velocity of the convective motion—far from the dynamo onset—would favor the high-order harmonics of the magnetic field even more.

Based on this analysis and with the estimated properties of the silicates mentionned above, to have a dominant dipolar component, one would need to have a rotation period of the planet shorter than 2 days. Such a short period is incompatible with a tidally locked planet in the habitable zone of any star but could be achieved for non-tidally locked planets. Magma oceans on tidally locked Super Earths are thus likely to generate multipolar and not dipolar magnetic fields.

We have to stress here that the scaling laws were extrapolated out of the parameter space accessible by dynamo simulations. For instance, the minimum Ekman number investigated was $10^{-6}$ (ref. [51]), while in our case we have $E \sim 10^{-12}$. We also considered a dynamo in a magma ocean solely. However, if the planet is differentiated, it is possible that it has a liquid iron core which could also generate a magnetic field. The interaction between the liquid core magnetic field and the magma ocean is not easy to predict but could result in a significant—or even dominant—dipolar component.

Over all, our simulation results show that silicates electric conductivity significantly increases when they melt at the megabar pressures. The conductivity in the liquid is a factor of 2 to 10 higher than the one in the solid depending on the material. This is because disorder favors band gap closure, and thus the gap closes as the system melts creating a semimetal while the solid phase under similar conditions remains a semiconductor. The calculation of the thermal conductivity shows a significant deviation from the Wiedmann–Franz law, which underestimates the thermal conductivity by a factor of 2 to 3 for a given electric conducitivity. For the three silicate materials considered here, the reflectivity in the solid phase is zero while the liquid exhibits an appreciable reflectivity. This difference in reflectivity could be a useful tool to help identifying the melting curve of the silicates. However, the low values of reflectivity and the persistence of an equilibrium between solid and liquid phase over a wide range of parameters might hinder a detection.

The values of the conductivity we obtained, in the range of a few hundreds of $\Omega^{-1}$ cm$^{-1}$, suggest a magnetic diffusivity of 16 $-40$ m$^2$/s, which could be compatible with a dynamo process

within the mantle of Super-Earths if the convective velocity is higher than $0.6-1.6$ mm/s. The exact value for the dynamo onset strongly depends on the conductivity which is sensitive to the composition. The latter is however very poorly constrained for Super Earth and may vary significantly from one system to another. However, we anticipate the conductivity of silicates to remain in the range of a few hundreds of $\Omega^{-1}$ cm$^{-1}$ as MgO, SiO$_2$, and MgSiO$_3$ all stay within this range. Nevertheless, the addition of metallic elements such as iron or nickel could significantly increase the value of the electric conductivity[54].

Based on dynamo models and order of magnitude estimates of the local Rossby number, we predict that, for Super-Earths with a rotation period longer than 2 days, a magma ocean is likely to generate a multipolar field and not a dipole dominated field, which could make a remote detection more difficult. On the other hand, if the planet is differentiated and a liquid iron core exists, the convection and magnetic field generation may be much more efficient in the core than in the mantle. The interaction of the magnetic field generated by the core and the conducting magma ocean makes a prediction for the actual field very challenging. It is thus of importance to characterize the behavior of the deep interior of the Super-Earths to fully understand how the different elements are distributed, the energy is transferred, and the magnetic fields generated.

## Methods

**Molecular dynamics simulations.** We present here results from molecular dynamics (MD) simulations coupled with density functional theory (DFT). We used the Vienna ab initio simulation package (VASP)[55] to perform the DFT-MD simulations at constant density and temperature, employing a Nosé thermostat[56,57]. We used a time step of 0.5–0.8 fs for a total duration of at least 1 ps. The simulation cell contained from 60 to 144 atoms depending on the material and the phase (see the details in Supplementary Table 1). The cell was given the appropriate shape for a given crystallographic symmetry or a simple cubic shape for liquids. We applied periodic boundary conditions. The DFT calculations were performed using the finite temperature Kohn–Sham scheme[58,59]. We employed the Perdew, Burke and Ernzerhof (PBE) generalized gradient approximation (GGA) exchange and correlation functional[60]. We used projector augmented waves (PAW) pseudo-potentials[61] which included frozen cores: $1s^2$ for oxygen, $1s^2 2s^2$ for magnesium, and $1s^2 2s^2 2p^6$ for silicon. We sampled the Brillouin zone with up to $2^3$ Monkhorst–Pack k-points grid[62] for the solid, and the $\Gamma$-point for liquid phases (see the details in Supplementary Table 1). We adjusted the energy cut-off so as to have convergence of the pressure and the internal energy to better than the statistical uncertainty of our MD simulations. The number of bands was adjusted so that the full spectrum of occupied and partially occupied states is reproduced.

We checked that there was no finite-size effects on the results by computing a few larger simulations. As can be seen in Supplementary Figures 1 and 2, 64- and 120-atom simulations of MgO give very similar results regarding the time evolution of the thermodynamic quantities. We also observe that the equilibrium is achieved with a stationary state. At an atomic level we also checked that we achieved two distinct states by looking at the mean-squared displacement of the different types of nuclei (see Supplementary Figures 3 and 4). In the solid the atoms remain at the same sites during the whole duration of the simulation. For the liquid phase, we observe a constant diffusion without getting locked in a glassy state.

**Conductivity and optical properties.** We calculated the electronic conductivity using a Kubo–Greenwood formula[63]. The optical properties are then computed using a Kramers–Kronig formula. Because these calculations are highly demanding, we followed a similar protocol as ref. [64]. Namely, we extracted the ionic positions from the DFT-MD simulations every 500 time steps. We then recomputed the electronic structure using the Abinit code[65], with a slightly higher energy cut-off (50 Ha) up to a $4^3$ Monkhorst–Pack k-points grid (see the details in Supplementary Table 1) and additional bands to be able to determine the conductivity up to an excitation energy of 1 Ha. Because the finite size of the cell limits the resolution in the eigen energies, the DC conductivity is not directly computed. We used instead a common method consisting in extrapolating a linear regression at low excitation energy and used this value for the DC conductivity. We do not expect any large changes in the conductivity behavior at very low energy. In Supplementary Figure 5, we plotted an example of conductivity calculation for MgO at 470 GPa and 12,000 K. The individual calculations fluctuate significantly indicating how strongly the microscopic configuration influences the properties. We also compared the 64-atom simulation with the 120-atom results. We see a slight increase in the low-frequency conductivity when the number of atoms increases, which indicates a marginal finite-size effects. It stays within the range of

fluctuations and the difference is not significant to have an impact on our conclusions. We further see that for MgO, the drop in conductivity at low frequency happens at even lower frequency for the larger cell. The drop is thus non physical but rather an artefact of the finite size of the cell and the limitation on the resolution of the bands for single snapshots.

The electronic contribution to the thermal conductivity was also computed within the linear response theory framework, using a Chester–Tellung–Kubo–Greenwood formula of the Onsager coefficients[63]. The snapshot DFT recalculations used for the electric conductivity estimation were also used for the thermal conductivity computation. We averaged the frequency-dependent conductivity as a function of time. We extrapolated the value to zero frequency the same way we determined the DC electric conductivity, with an extrapolation from the low frequency values.

In order to verify our results, we also determined the DOS of MgO using the AM05 exchange-correlation functional[66] that has been already used for MgO[15]. We observed some very small differences in the deep bound states and in the very excited states, but the structure of the DOS using AM05 or PBE is identical near the Fermi level, which suggests that the conductivity is very similar for PBE and AM05. It is however known that regular GGA functionals tend to understimate the gap energy in insulators. Hybrid functionals have proven to be more accurate for insulators at zero temperature[67]. To estimate the influence of including an Hartree–Fock correction we reused the snapshots from the PBE trajectory and recomputed the DOS of liquid and B2 MgO at 470 GPa and 120,000 K, using PBE-based PAW pseudo-potentials and the HSE06 functional for the self-consistent loop calculation. We used VASP for technical reasons. The results are plotted in Supplementary Figures 6 and 7. It is clear that the hybrid functional tends to slightly open the gap, by about 1 eV for the solid phase. The metallicity of the liquid is also slightly lower with HSE than with PBE. However, qualitatively, the use of an HF correction does not change the properties of the material and we still observe a semiconductor as the B2 phase and a semi-metallic liquid. In order to estimate the influence of the HSE functional on the conductivity itself, we performed an additional calculation of the conductivity for liquid MgO as can be seen in Supplementary Figure 8 using VASP. We made sure that both VASP and Abinit produced similar results with PBE. When we use HSE06, the conductivity is qualitatively similar with a shift in the energy at the conductivity maximum due to the opening of the gap. The DC conductivity decreases slightly when switching from PBE to HSE, but only by 36%. It is very likely that the actual value lies in between the HSE06 and the PBE predictions but this does not change the order of magnitude nor the conclusion regarding the magnetic field generation.

Using the calculated conductivity and a Kramers–Kronig we computed the optical index and the reflectivity[63] of the different silicates. For the reflectivity, it is necessary to know the properties of the unshocked material. We used the following values of the optical index for the unshocked phase: $n_0 = 1.74$ for MgO[68], $n_0 = 1.799$ for the stishovite[33], and $n_0 = 1.66$ for MgSiO$_3$ enstatite[69].

**Code availability**. VASP is a proprietary software very commonly used. Abinit is an open-source code publicly available. All the scripts that we used to produce the different graphs are available upon request to the authors.

## Data availability

The simulations setup and results are available upon request. They are not publicly available because of the large size of the output files.

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

# ARTICLE

45. Ohta, K. Lattice thermal conductivity of MgSiO₃ perovskite and post-perovskite at the core-mantle boundary. *Earth Planet. Sci. Lett.* **349–350**, 109–115 (2012).

46. Konôpková, Z. et al. Direct measurement of thermal conductivity in solid iron at planetary core conditions. *Nature* **534**, 99–101 (2016).

47. Ohta, K. et al. Experimental determination of the electrical resistivity of iron at Earth's core conditions. *Nature* **534**, 95–98 (2016).

48. Militzer, B. Ab initio investigation of a possible liquid-liquid phase transition in MgSiO₃ at megabar pressures. *High Energy Dens. Phys.* **9**, 152–157 (2013).

49. Militzer, B. et al. Understanding Jupiter's Interior. *J. Geophys. Res. Planets* **121**, 1552 (2016).

50. Frazer, M. C. Temperature gradients and the convective velocity in the Earth's core. *Geophys. J.* **34**, 193–201 (1973).

51. Christensen, U. R. Dynamo scaling laws and applications to the planets. *Space Sci. Rev.* **152**, 565–590 (2010).

52. Elkins-Tanton, L. T. et al. Early magnetic field and magmatic activity on Mars from magma ocean cumulate overturn. *Earth. Planet. Sci. Lett.* **236**, 1–12 (2005).

53. Kopparapu, R. K. et al. Habitable zones around main-sequence stars: new estimates. *Astrophys. J.* **765**, 131 (2013).

54. Holmström, E. et al. Electronic conductivity of solid and liquid (Mg, Fe)O computed from first principles. *Earth. Planet. Sci. Lett.* **490**, 11–19 (2018).

55. Kresse, G. & Furthmüller, J. Efficient iterative schemes for ab initio total-energy calculations using a plane-wave basis set. *Phys. Rev. B* **54**, 11169–11186 (1996).

56. Nosé, S. A unified formulation of the constant temperature molecular dynamics methods. *J. Chem. Phys.* **81**, 511–519 (1984).

57. Nosé, S. Constant temperature molecular dynamics methods. *Prog. Theor. Phys. Suppl.* **103**, 1 (1991).

58. Mermin, N. D. Thermal properties of the inhomogeneus electron gas. *Phys. Rev.* **137**, 1441 (1965).

59. Kohn, W. & Sham, L. J. Self-consistent equations including exchange and correlation effects. *Phys. Rev.* **140**, 1133 (1965).

60. Perdew, J. P., Burke, K. & Ernzerhof, M. Generalized gradient approximation made simple. *Phys. Rev. Lett.* **77**, 3865–3868 (1996).

61. Blöchl, P. E. Projector augmented-wave method. *Phys. Rev. B* **50**, 17953–17979 (1994).

62. Monkhorst, H. J. & Pack, J. D. Special points for Brillouin-zone integrations. *Phys. Rev. B* **13**, 5188 (1976).

63. Mazevet, S. et al. Calculations of the transport properties within the PAW formalism. *High. Energy Dens. Phys.* **6**, 84–88 (2010).

64. Soubiran, F. et al. Optical signature of hydrogen-helium demixing at extreme density-temperature conditions. *Phys. Rev. B* **87**, 165114 (2013).

65. Gonze, X. et al. Abinit: First-principles approach to material and nanosystem properties. *Comput. Phys. Commun.* **180**, 2582–2615 (2009).

66. Armiento, R. & Mattsson, A. E. Functional designed to include surface effects in self-consistent density functional theory. *Phys. Rev. B* **72**, 085108 (2005).

67. Schimka, L., Harl, J. & Kresse, G. Improved hybrid functional for solids: The HSEsol functional. *J. Chem. Phys.* **134**, 024116 (2011).

68. Stephens, R. & Malitson, I. Index of refraction of magnesium oxide. *J. Res. Natl. Bur. Stand.* **49**, 249 (1952).

69. Anthony, J. W., et al. Enstatite. In Handbook of Mineralogy (eds. Anthony, J. W. et al.) (Mineralogical Society of America, Chantilly, 2001).

## Acknowledgements

We thank D. Fratanduono, A. Ravasio, R. Bolis, and M. Millot for the insightful discussions about the experimental measurements. We are grateful to B. Buffet for his explanation of the magnetic field generation in the Earth. The present work was supported by the U.S. National Science Foundation (AST 1412646) and in part by the U.S. Department of Energy (DE-SC0016248). We used the high-performance computing facilities Comet-SDSC and Stampede-TACC from the XSEDE program and Pleiades from NAS-NASA.

## Author contributions

On an original idea of B.M., F.S. and B.M. designed the research. F.S. performed most of the simulations, analyzed and interpreted the results. F.S. and B.M. wrote the manuscript.

## Additional information

**Competing interests:** The authors declare no competing interests.

