## [Peer Review File · Nature Communications]

Reviewers' comments:

Reviewer #1 (Remarks to the Author):

The authors present results of DFT-MD simulations for the EOS and the electrical and thermal conductivity of MgO, SiO₂, and MgSiO₃. These materials are probably the main constituents in the interior of Earth and of super-Earths (SE). The manuscript contributes to the discussion of the high-pressure phase diagram by adding predictions for the Hugoniot curve (Fig. 1) which follow the trends of the experiments and of DFT-MD simulations performed earlier. The main part of the paper is devoted to the electronic and transport properties of these materials. The authors present results for the DOS (Fig. 2), the electrical conductivity (Fig. 3, electronic contribution), estimates for the thermal conductivity, and data for the reflectivity (Fig. 4). They discuss the possible impact of their results on the magnetic field generation in SE in the last chapter.

While the results for the EOS and the Hugoniot curve are of some interest for the still ongoing discussion of the high-pressure phase diagram of these minerals, the results for the electronic and transport properties are disputable (strictly speaking: they are wrong) and the corresponding conclusions are highly speculative. The authors study the changes in the electronic properties in these materials along the solid-liquid phase transition and find band gap closure, i.e. a transition from an insulating or semi-conducting state to a semi-metal behavior. The critical quantity in these calculations is the band gap. In DFT it is known for a long time that the PBE XC functional used by the authors (they have tested the AM05 XC functional and find similar results) underestimates the band gap systematically by up to 50-60%. Therefore, PBE gives wrong predictions for the electronic transitions mentioned above and, in particular, also for the electronic and thermal conductivity and the reflectivity. The results are thus misleading. The discussion of the magnetic field generation in SE given in the last chapter is highly speculative anyway. Taking into account that the estimates for the conductivities and related quantities like magnetic diffusivity and magnetic Reynolds number are based on wrong numbers (the respective values can change easily by a factor of two up to an order of magnitude if calculated correctly), the conclusions drawn by the authors are unfounded.

This fundamental problem can only be addressed properly by using, e.g., hybrid functionals like HSE or exact-exchange treatments within LDA (EXX-LDA). The at least partially exact treatment of exchange in such DFT-MD simulations leads to a drastic increase of the computational costs but, simultaneously, to well-founded results. This has been shown in other studies on the behavior of materials under extreme conditions. Furthermore, the authors disclose only few details of their evaluation of the Kubo-Greenwood formula. In particular I miss more detailed comments on convergence checks with respect to the particle number, cut-off energy, and the use of pseudo-potentials, more than given in the 'methodology' section. Based on the sparse information given I doubt that the extrapolations to the DC conductivity shown in Fig. 3 are correct. The calculation of the values for the thermal conductivity remains unclear so that the discussions with respect to the deviations from the Wiedemann-Franz law are dubious.

I conclude that the paper contains in large parts misleading results and that the corresponding conclusions are highly speculative, if not wrong. The paper should be rejected.

If the authors aim at submitting a considerably revised version of their manuscript with a proper treatment of the band-gap problem to another journal, I have additional comments which might be helpful:

1. The paper would gain from a more critical description of the methods used and the results obtained. In particular, the presentations in the abstract and in the 'magma ocean' section are exaggerated.
2. A comprehensive review on extra-solar planets has been given recently by H. Rauer et al. (2014) which could be referred to in the introduction.
3. The discussion of band-gap closure could gain from a more detailed analysis. For instance, the

data shown rather imply the formation of a pseudo-gap than band-gap closure. The problem of insulator-to-metal transitions has been in the focus of physical research since decades so that it might be useful to inspect basic work of Mott and others (e.g., P. P. Edwards) in this context.

4. Currently, there is a lively discussion on the the electrical and thermal conductivity at the CMB of Earth. Besides Ohta et al. (Ref. 45) Konopkova et al. (published in the same issue of Nature!) found contradicting results. The authors should address this issue in their discussions.

Reviewer #2 (Remarks to the Author):

The manuscript 'Electrical Conductivity and magnetic dynamos in magma oceans of Super-Earths' deals with the importance of understanding the influence of the magma ocean on the polarity of magnetic field of Super-Earth exoplanets. For such planets, the boundary between the magnetic field and the solar wind (the magnetopause) tends to be closer to the surface, making a remote detection more challenging.

By means of molecular dynamics and optical as well as electronic properties calculations within the ab-initio framework, the authors have been able to predict not only that the electron conductivity increases when silicates melt at the megabar of pressure but also that the magma ocean on tidally locked Super Earths are likely to generate multipolar magnetic fields.

The introduction of the manuscript provides a good and generalised background of the topic that quickly gives the reader an appreciation of the problem under study. The authors also provided an adequate number of references to substantiate the study.

The reasoning about how the materials were chosen for the present study and the methodology adopted to investigate the problem are also appropriate.

The objective is clearly defined in the last paragraph of the introduction.

The manuscript is well written and generally easy to read, even for researchers not strictly related to Earth and Planetary sciences and ab-initio calculations. The main body of the paper counts four sections that successfully guide the reader from the early stages where the models were chosen to the final calculations done to reach the aim of the study. The figures are readable despite the complexity that sometimes they can reach, their number adequate to the manuscript and their captions clear in explaining the content of the panels. The overall set of references is more than enough to support the narrative. The analysis of the data is appropriate and the conclusions are well supported by the results.

The authors have demonstrated for the first time, by means of ab-initio calculations, that:

1) for the three materials considered in the present study, it is possible to use the reflectivity in experiments as a useful tool to help identify melting curves in silicates;

2) from the values of the calculated conductivity, magnetic diffusivity and the convective velocity of the magma ocean, it is possible to infer that for planets with a rotation period longer than 2 days, a magma ocean is likely to generate a multipolar magnetic field;

3) a prediction for interaction magnetic field coming from magma ocean and differentiated liquid iron core remains very challenging

For the novelty of the results and the overall construct of the manuscript I would recommend this paper for publication in Nature Communication once the authors have addressed the following questions:

1) In the 4th line of the second paragraph of Section 'Electronic gap closure upon melting', it is stated that the energy gaps are in the '...order of 2 to 4 eV'. In the panels of figure 2 I cannot find band gaps bigger than 3 eV. Moreover, the plateau on the conductivity (blue curves) representing the energy gap of the semi-conductor in figure 3 seems to be bigger than the width for the blue region in figure 2 with the only exception of MgSiO₃ PPV where the two figures agree. Could you please explain this mismatch?

2) Talking about band gaps in the solid, the authors stated in the methodology that the calculations were done on a 2x2x2 k-point grid. Did the authors calculate the electronic band structures for the three materials? If so, would it be possible to have some graphs maybe shown in a supplementary document? How dense is the valence band region for the solid? Any particular shape in the conduction bands to justify the same conductivity at higher energy between solid and liquid materials?

3) In the methodology section 'Conductivity and optical properties', the authors justify the overall use of the PBE functional by affirming that only very small differences were found in the Density of States between PBE and AM05 functionals. It is known that AM05 is a good descriptor and that its precision can be compared to much more expensive (in terms of CPU time) hybrid functionals (i.e., HSE06). This has been proven to be true when describing structural properties of solids and surfaces such as lattice parameters and bulk modulus. Fewer studies were conducted using different functionals to compare the electronic properties of materials and in these few cases, everything seems to specifically depend on the kind of material. Did the authors test AM05 against for example HSE06 or other hybrid calculations of both Density of States and band structures?

4) The authors used two different ab-initio codes: VASP mainly for the molecular dynamics simulation and Abinit to obtain an estimate of the conductivity and the optical properties of the above mentioned materials. Did they check that the two codes give similar results in terms of energetics and structural and electronic properties?

5) In the time frame of the molecular dynamics simulations, did the authors check the self-diffusion coefficients of the atomic components in particular at very high pressure to ensure that the materials did not undergo phase transition (i.e., liquid-glass phase transition)?

Finally, even though the amount of data and setup for such calculations is considerable, it would be really appreciated if at least some material could be included in an additional document as Supporting material to help the reader to fully understand and easily reproduce the data presented in the manuscript.

Reviewer #3 (Remarks to the Author):

The authors conduct three numerical simulations to measure the phase, electrical conductivity, and thermal conductivity at high P-T. They choose a single P-T state for MgO, SiO₂ and MgSiO₃. They find partial band gap closure indicating semi-metallic behavior.

On page 3, the authors state that they find the Lorenz number for these materials is factor of 2-3 larger than for metals. But do they find that the conductivities are linearly related at all? Or are these Lorenz numbers from single data points? Could the conductivities be related in some other way (power law, etc)?

For a magma ocean with Mg, O, and Si, what abundances of the species (MgO, SiO₂, MgSiO₃) do you expect? If its a mixture then how would you estimate the conductivity of the mixture? In practice would the high or low conductivity species dominate?

Page 5, top left: This is strange. You assume $R_m=40$ (onset of dynamo) to get velocity. Then you use that onset velocity to estimate Ro_I . But at onset Ro_I should be small, less than the dipolar value of 0.1. You can't both be at dynamo onset and multipolar.

We appreciated the constructive feedback from the three referees and have revised our manuscript accordingly. Most importantly, we performed additional calculations with the HSE06 functional in order to address the criticism of the referee #1 who did not trust our results because they were derived with the PBE functional, which is known to underestimate band gaps of insulators and semi-conductors at zero temperature. We have thus repeated this core part of conductivity calculation for MgO with the HSE06 functional and essentially reproduced our earlier PBE predictions. With either functional, we see a significant difference in the band gap between the solid and liquid on the melting line. This difference is a robust result that does not depend on the choice of functional. The difference in the band gap is caused by increased disorder in the liquid phase. The subsequent predictions in our manuscript were not affected by the choice of functional either. While it took some time to perform the HSE calculations, in our view, we now have sufficiently addressed the criticism of the referee #1. In the following text, we present some additional arguments and respond in detail to the comments of all three referees.

Reviewer #1 (Remarks to the Author):

The authors present results of DFT-MD simulations for the EOS and the electrical and thermal conductivity of MgO, SiO₂, and MgSiO₃. These materials are probably the main constituents in the interior of Earth and of super-Earths (SE). The manuscript contributes to the discussion of the high-pressure phase diagram by adding predictions for the Hugoniot curve (Fig. 1) which follow the trends of the experiments and of DFT-MD simulations performed earlier. The main part of the paper is devoted to the electronic and transport properties of these materials. The authors present results for the DOS (Fig. 2), the electrical conductivity (Fig. 3, electronic contribution), estimates for the thermal conductivity, and data for the reflectivity (Fig. 4). They discuss the possible impact of their results on the magnetic field generation in SE in the last chapter.

While the results for the EOS and the Hugoniot curve are of some interest for the still ongoing discussion of the high-pressure phase diagram of these minerals, the results for the electronic and transport properties are disputable (strictly speaking: they are wrong) and the corresponding conclusions are highly speculative. The authors study the changes in the electronic properties in these

materials along the solid-liquid phase transition and find band gap closure, i.e. a transition from an insulating or semi-conducting state to a semi-metal behavior. The critical quantity in these calculations is the band gap. In DFT it is known for a long time that the PBE XC functional used by the authors (they have tested the AM05 XC functional and find similar results) underestimates the band gap systematically by up to 50-60%. Therefore, PBE gives wrong predictions for the electronic transitions mentioned above and, in particular, also for the electronic and thermal conductivity and the reflectivity. The results are thus misleading. The discussion of the magnetic field generation in SE given in the last chapter is highly speculative anyway. Taking into account that the estimates for the conductivities and related quantities like magnetic diffusivity and magnetic Reynolds number are based on wrong numbers (the respective values can change easily by a factor of two up to an order of magnitude if calculated correctly), the conclusions drawn by the authors are unfounded.

This fundamental problem can only be addressed properly by using, e.g., hybrid functionals like HSE or exact-exchange treatments within LDA (EXX-LDA). The at least partially exact treatment of exchange in such DFT-MD simulations leads to a drastic increase of the computational costs but, simultaneously, to well-founded results. This has been shown in other studies on the behavior of materials under extreme conditions. Furthermore, the authors disclose only few details of their evaluation of the Kubo-Greenwood formula. In particular I miss more detailed comments on convergence checks with respect to the particle number, cut-off energy, and the use of pseudo-potentials, more than given in the 'methodology' section. Based on the sparse information given I doubt that the extrapolations to the DC conductivity shown in Fig. 3 are correct. The calculation of the values for the thermal conductivity remains unclear so that the discussions with respect to the deviations from the Wiedemann-Franz law are dubious.

I conclude that the paper contains in large parts misleading results and that the corresponding conclusions are highly speculative, if not wrong. The paper should be rejected.

We have heard the strong criticism from the referee and have given a lot of consideration to his statements on the inadequacy of PBE for the determination of accurate electronic properties. The choice of an exchange-correlation functional in lieu of another can be critical on numerous aspect. And PBE as well

as most of the GGA-type functionals are known to inherently underestimate the band gaps of insulators and semi-conductors at zero temperature. In our view, the magnitude of the discrepancy – 50-60% – quoted by the referee is however rather large but unfortunately close to the truth on the most extreme cases (see for instance Schimka et al., JCP 134, 024116 (2011)). We also want to remind the referee that experiments are also subject to caution since deviations of up to 1 eV are not uncommon for different experimental set-ups (see e.g. the compiled values reported in Ataide et al., PRB 95, 045126 (2017)). We agree with the referee that on this specific aspect of the band gap calculation, the use of more sophisticated generalized Kohn-Sham schemes including Hartree-Fock-like calculations can improve the accuracy. It has however already been shown that the correction induced by GW types of calculation becomes less significant as temperature increases (see Faleev et al., PRB 74, 033101 (2006))

Since the referee suggests the computations should have been performed with the HSE functional instead of PBE, we have repeated the core part of the conductivity calculation for MgO close to the melting line with this functional and essentially reproduced our earlier PBE predictions.

For several snapshots we recomputed the DOS with the HSE06 functional. We had technical difficulties to perform the calculation on Abinit and opted for a calculation using VASP. We include two figures R1 and R2 in the rebuttal. R2 is an enlarge section of figure R1. We found a difference between PBE and HSE06 but the magnitude is very small. There is still a large difference between the solid and the liquid and, thus, qualitatively the conclusions of our article remain the same. With HSE06, the gap is slightly wider in the solid, but as for PBE, the gap is closed in the liquid.

Due to technical difficulties, we were not able to compute the conductivity with the HSE06 functional. Nevertheless, the integration of the DOS above the Fermi energy indicates that the electronic density in the continuum is 55% lower for the solid and 32% lower in the liquid if HSE results are compared to PBE predictions. To first order, the electrical conductivity is proportional to the electronic density in the continuum. This means that in the liquid the conductivity is approximately 32% lower with HSE06 functional than with PBE. This is still within the range of conductivities we used for the estimation on the magma ocean field generation. But even using this decrease of 32%, we obtain a minimum convective velocity increased by 50% and we obtain a local Rossby number increase of 60%. None of our conclusions would be impacted by such a difference except the exact

values at the onsets.

R 1: Density of state of MgO at 470 GPa and 12000 K in the B2 and in the liquid phases, using PBE or HSE06. The full lines are the full DOSs, the dashed lines represent the occupied DOSs.

R 2: Zoom on the density of state of MgO at 470 GPa and 12000 K in the B2 and in the liquid phases, using PBE or HSE06. The full lines are the full DOSs, the dashed lines represent the occupied DOSs.

We added a paragraph in the method section to mention the comparison with HSE06 after the AM05 discussion and we added a more thorough discussion in the Supplementary Material.

If the authors aim at submitting a considerably revised version of their manuscript with a proper treatment of the band-gap problem to another journal, I have additional comments which might be helpful:

1. The paper would gain from a more critical description of the methods used and the results obtained. In particular, the presentations in the abstract and in the 'magma ocean' section are exaggerated.

We hope that our reply regarding the hybrid functional and the additional results using HSE06 will convince the referee to trust the predictions of our work. Nevertheless we slightly modified the tone in the abstract and the "Magma ocean" section.

2. A comprehensive review on extra-solar planets has been given recently by H. Rauer et al. (2014) which could be referred to in the introduction.

We thank the referee for pointing out this review. We included it in the

introduction.

3. The discussion of band-gap closure could gain from a more detailed analysis. For instance, the data shown rather imply the formation of a pseudo-gap than band-gap closure. The problem of insulator-to-metal transitions has been in the focus of physical research since decades so that it might be useful to inspect basic work of Mott and others (e.g., P. P. Edwards) in this context.

We agree with the referee that the exact mechanism for the band-gap closure is unclear and it might very well be through the formation of a pseudo-gap since the band-gap might still be open for some of the K-points. Because it is way beyond the aim of this work we did not investigate the complete electronic structure. We also want to stress that the conductivity calculation in the linear perturbation theory framework is blind to the exact gap closure mechanisms as long as it is well converged in K-points, cut-offs, number of particles and bands. The details of the “gap closure” would be very interesting to investigate to clearly understand what makes the solid and the liquid so different. A study such as the one performed by Naumov & Hemley, PRL 114, 156403 (2015) using Wannier centers and group theory, would be definitely interesting. But this is beyond the scope of the present article, which is rather to identify a process (increase of the conductivity at the melting) that has consequences for planets (possibility for an intra-mantle dynamo).

4. Currently, there is a lively discussion on the the electrical and thermal conductivity at the CMB of Earth. Besides Ohta et al. (Ref. 45) Konopkova et al. (published in the same issue of Nature!) found contradicting results. The authors should address this issue in their discussions.

We accidentally omitted to refer to the second experimental results. We added the reference and a very brief comment in the 3rd paragraph of the “Electronic transport properties upon melting” section:

Under the same conditions, the thermal conductivity of iron has been measured between 33 W/m/K [46] and 226 W/m/K [47]. The discrepancy in the thermal conductivity measurements underlines the difficulty of such experiments and the still large uncertainty on the iron conductivity in Earth’ core.

Reviewer #2 (Remarks to the Author):

The manuscript 'Electrical Conductivity and magnetic dynamos in magma oceans of Super-Earths' deals with the importance of understanding the influence of the magma ocean on the polarity of magnetic field of Super-Earth exoplanets. For such planets, the boundary between the magnetic field and the solar wind (the magnetopause) tends to be closer to the surface, making a remote detection more challenging.

By means of molecular dynamics and *ab-initio* optical as well as electronic properties calculations within the *ab-initio* framework, the authors have been able to predict not only that the electron conductivity increases when silicates melt at the megabar of pressure but also that the magma ocean on tidally locked Super Earths are likely to generate multipolar magnetic fields.

The introduction of the manuscript provides a good and generalised background of the topic that quickly gives the reader an appreciation of the problem under study. The authors also provided an adequate number of references to substantiate the study.

The reasoning about how the materials were chosen for the present study and the methodology adopted to investigate the problem are also appropriate. The objective is clearly defined in the last paragraph of the introduction.

The manuscript is well written and generally easy to read, even for researchers not strictly related to Earth and Planetary sciences and *ab-initio* calculations. The main body of the paper counts four sections that successfully guide the reader from the early stages where the models were chosen to the final calculations done to reach the aim of the study. The figures are readable despite the complexity that sometimes they can reach, their number adequate to the manuscript and their captions clear in explaining the content of the panels. The overall set of references is more than enough to support the narrative. The analysis of the data is appropriate and the conclusions are well supported by the results.

The authors have demonstrated for the first time, by means of *ab-initio* calculations, that:

- 1) for the three materials considered in the present study, it is possible to use the reflectivity in experiments as a useful tool to help identify melting curves in silicates;
- 2) from the values of the calculated conductivity, magnetic diffusivity and the convective velocity of the magma ocean, it is possible to infer that for planets with a rotation period longer than 2 days, a magma ocean is likely to generate a multipolar magnetic field;
- 3) a prediction for interaction magnetic field coming from magma ocean and differentiated liquid iron core remains very challenging

For the novelty of the results and the overall construct of the manuscript I would recommend this paper for publication in Nature Communication once the authors have addressed the following questions:

We are very grateful to the referee for his positive review. We are also very glad that our take home messages were clear enough to be identified and summed up by the referee.

1) In the 4th line of the second paragraph of Section 'Electronic gap closure upon melting', it is stated that the energy gaps are in the '...order of 2 to 4 eV'. In the panels of figure 2 I cannot find band gaps bigger than 3 eV. Moreover, the plateau on the conductivity (blue curves) representing the energy gap of the semiconductor in figure 3 seems to be bigger than the width for the blue region in figure 2 with the only exception of MgSiO₃ PPV where the two figures agree. Could you please explain this mismatch?

There are a few different explanations regarding the mismatch. The first thing, is that we have to remind the referee that the DOSs that are plotted are averaged over time, with a smearing (0,2 eV) and over the K-points. The colored region on the graphs of Fig. 2 is a very conservative minimum value, but for instance, on MgO, a value of 4 eV is more likely (between -2 and 2 eV on the plot, where you see the "walls" of the gap). Now the gap on the DOS is either direct or indirect and it is more likely indirect especially on MgO and SiO₂. However, for the

purpose of the conductivity calculations, only the transitions at the same K-point are considered and they are blind to indirect gaps. Therefore, if there is an indirect gap of 4 eV but direct gaps are all larger than 5 eV, the plateau in the conductivity is going to be of 5 eV. This is a limitation of the linear response theory.

2) Talking about band gaps in the solid, the authors stated in the methodology that the calculations were done on a 2x2x2 k-point grid. Did the authors calculate the electronic band structures for the three materials? If so, would it be possible to have some graphs maybe shown in a supplementary document? How dense is the valence band region for the solid? Any particular shape in the conduction bands to justify the same conductivity at higher energy between solid and liquid materials?

For the purpose of this article, we only computed the average DOS but we did not investigate the very detailed electronic structure K-point by K-point because it required to redo the already very expensive calculations.

Nevertheless, based on the DOS shown in Fig. R1, we would argue that the similarity in high energy conductivity is due to the similarity in the first block of valence states right below the Fermi energy. This shell of hybridized state is very similar for both the solid and the liquid phases. Since they also have a somewhat similar continuum, the conductivity is the same for the two systems. The small modulation of the lower energy section of the solid phase continuum, likely due to correlations between atomic sites, is the explanation for the slightly higher conductivity at high energy for the solid.

3) In the methodology section 'Conductivity and optical properties', the authors justify the overall use of the PBE functional by affirming that only very small differences were found in the Density of States between PBE and AM05 functionals. It is known that AM05 is a good descriptor and that its precision can be compared to much more expensive (in terms of CPU time) hybrid functionals (i.e., HSE06). This has been proven to be true when describing structural properties of solids and surfaces such as lattice parameters and bulk modulus. Fewer studies were conducted using different functionals to compare the electronic properties of materials and in these few cases, everything seems to specifically depend on the kind of material. Did the authors test AM05 against for example HSE06 or other hybrid calculations of both Density of States and band

structures?

We performed calculations with the HSE06 functional and found results that are very similar to our PBE predictions. Please see our reply to the first referee.

4) The authors used two different ab-initio codes: VASP mainly for the molecular dynamics simulation and Abinit to obtain an estimate of the conductivity and the optical properties of the above mentioned materials. Did they check that the two codes give similar results in terms of energetics and structural and electronic properties?

We have to say that we did not go into such details. Abinit and VASP are two codes that have been intensively used by many communities and give very consistent results. We used these two codes for two distinct purposes and have no reason to believe there could be any significant deviations between the two as long as we are considering accurate PAW pseudo-potentials and converged simulations, which we believe we did. Furthermore, the DOS obtained with VASP and with PBE for MgO seem very compatible.

5) In the time frame of the molecular dynamics simulations, did the authors check the self-diffusion coefficients of the atomic components in particular at very high pressure to ensure that the materials did not undergo phase transition (i.e., liquid-glass phase transition)?

We did not compute self-diffusion coefficients per se, but we carefully observed the mean-squared displacement as function of time and inspected movies illustrated the motion of the atoms. This way, we confirmed that the liquid simulation remained liquid with nuclei traveling throughout the box. The solid remained frozen with nuclei oscillating around their lattice sites. To show this behavior, we included two new graphs in the Supplementary Material.

Finally, even though the amount of data and setup for such calculations is considerable, it would be really appreciated if at least some material could be included in an additional document as Supporting material to help the reader to fully understand and easily reproduce the data presented in the manuscript.

We agree and have created supplementary materials with complementary figures

that will help with the reproducibility of the predictions and make the evaluation of our work easier.

Reviewer #3 (Remarks to the Author):

We thank the referee for his review and took his comments into account. We offer some explanation to his questions below.

The authors conduct three numerical simulations to measure the phase, electrical conductivity, and thermal conductivity at high P-T. They choose a single P-T state for MgO, SiO₂ and MgSiO₃. They find partial band gap closure indicating semi-metallic behavior.

On page 3, the authors state that they find the Lorenz number for these materials is factor of 2-3 larger than for metals. But do they find that the conductivities are linearly related at all? Or are these Lorenz numbers from single data points? Could the conductivities be related in some other way (power law, etc)?

The reported values of the Lorenz numbers are from single data point. We unfortunately do not have enough sampling to derive an other law. It would be too speculative. It is very likely that a relationship exists between the electric conductivity and the electronic contribution of the thermal conductivity since they are associated to the same phenomenon. However, because the system is only semi metallic, the conductivity is heavily influenced by the dynamics of the nuclei and this might take a different form for the electric and the thermal conductivities. For the Wiedemann-Franz law to hold, the contribution from the lattice or the ionic movements has to be negligible. The point of our comment is to stress the discrepancy on the Lorenz numbers and to say that it is necessary to be cautious when using it to infer properties from the sole determination of one of the two conductivities. To that aim, we think that our examples are significant enough to be mentioned without having to explore in greater details the physics. That would require a much larger dataset of points and a systematic study of these points which is beyond the scope of this current project.

For a magma ocean with Mg, O, and Si, what abundances of the species (MgO, SiO₂, MgSiO₃) do you expect? If its a mixture then how would you estimate the conductivity of the mixture? In practice would the high or low conductivity species dominate?

As far as we know, any guess on the composition is as good as another. If we use the Earth as a proxy, we are probably at something close to 25 – 25 – 50 mass percent of Mg, Si and O which would be roughly the composition of perovskite (MgSiO₃). However, large variations in molecular clouds and in stellar atmospheres have been observed, which means that the bulk composition of extra-solar Super-Earth could be completely different. Nevertheless, the two end-members are MgO and SiO₂ and the conductivity is going to be some composite of the conductivities of these end-members. The exact value is difficult to infer since you have for instance MgSiO₃ being liquid under conditions for which MgO is solid and therefore has a low conductivity. But we anticipate the order of magnitudes to be roughly similar in the liquid phases of the different silicates. It is clear that the next steps will be to carefully constrain the effect of composition on the conductivities. We added a short comment in the conclusion:

“ The exact value for the dynamo onset strongly depends on the conductivity which is sensitive to the composition. The latter is however very poorly constrained for Super Earth and may vary significantly from one system to another. However, we anticipate the conductivity of silicates to remain in the range of a few hundreds of $\Omega^{-1} \text{ cm}^{-1}$ as MgO, SiO₂ and MgSiO₃ all stay within this range. Nevertheless, the addition of metallic elements such as iron or nickel could significantly increase the value of the electric conductivity.”

Page 5, top left: This is strange. You assume $Rm=40$ (onset of dynamo) to get velocity. Then you use that onset velocity to estimate Ro_l . But at onset Ro_l should be small, less than the dipolar value of 0.1. You can't both be at dynamo onset and multipolar.

Based on the geodynamo literature, it is unclear to us what the structure of the magnetic field is at the onset of the dynamo. While numerous examples of dipolar type dynamos close to the onset have been published in the recent years, it seems that multipolar fields, even close to the onset, are also possible (see for instance Christensen & Arnoud, Geophys. J. Int. (2006) 166, 97–114). The scaling laws seem to infer a strong effect due to the Coriolis force in shaping the geometry of the magnetic field, but this does not have much influence on the onset.

Nevertheless, we wanted to go for the most extreme case and take the lowest velocity for a dynamo to develop and see what the condition on the local Rossby

number would give. If we have a higher velocity, the local Rossby number will increase as the Rossby number is proportional to the velocity. That would result in a "more" multipolar field. We added a sentence in this paragraph:

"It is interesting to note that the local Rossby number is proportional to the convective velocity. This means that a higher velocity of the convective motion -- far from the dynamo onset -- would favor the high-order harmonics of the magnetic field even more."

Reviewers' comments:

Reviewer #1 (Remarks to the Author):

The authors have submitted a revised version that takes into account most of my suggestions. My main concern was related to the results for the electrical and thermal conductivity of MgO, SiO₂, and MgSiO₃. The PBE XC functional used in the DFT-MD simulations underestimates the band gap systematically so that conductivities or optical properties predicted on that basis, in particular along a path from an insulator to a semiconductor or bad metal as relevant for these materials and planetary interior conditions, are disputable.

The authors have followed my advise and evaluated the Kubo-Greenwood formula for a number of snapshots of the simulation using the HSE XC functional. The HSE calculations are, due the inclusion of parts of the non-local Hartree-Fock exchange, numerically much more expensive than the PBE calculations but predict the electronic band gap of many elements and compounds with much higher accuracy than the PBE XC functional can do. However, the authors were only able to extract the DOS and not the conductivities due to technical difficulties. Based on this rather limited new information they claim that the HSE results essentially reproduce the previous PBE results. I do not agree with this conclusion.

The HSE DOS shows clearly a wider band gap than PBE, see Figs. 6 and 7 in the SM. The occupation numbers and the dipole matrix elements entering the Kubo-Greenwood formula are influenced for those energies as well. Only a full HSE calculation would show the quantitative effect. The argument of the authors that the integrated DOS (this normalization gives the particle number or density) can serve as an estimate for the expected conductivity shifts does not apply in this case since only the conduction band is considered. This would give a good estimate for metallic states but not for semiconductors or semimetals as in the present case. A more detailed discussion on the conductivity should start from a proper transport theory and not from the DOS alone since disorder (via $S(k,\omega)$), screening (via $\epsilon(k,\omega)$) and collisions (ions, electrons, phonons) influence the conductivity. A simple Ziman theory where the DOS (or Fermi energy) plays the major role is not applicable for these conditions.

The authors repeatedly point out that the qualitative behavior, in particular the ratio of the conductivities for the solid and liquid phases, remains unchanged, regardless of using the PBE or the more suited HSE XC functional. It is clear that HSE would shift the conductivities of all materials for the interior conditions considered here to lower values with respect to PBE, both in the liquid and solid. The liquids would perhaps still have a higher dc conductivity than the solid phases. Better predictions for the transport properties would represent an important step forward in this context which, however, is not made in the paper.

Since the conductivities (but also viscosities and the density and temperature profiles enter the dynamo simulations) are still not known accurately, performing a parameter study would be a reasonable alternative so that the values for which a dynamo in the magma ocean actuates could be isolated. Interestingly, such a parameter study has already been done for the young Earth, see Ziegler and Stegman, *Geochem. Geophys. Geosyst.* 14, 4735 (2013), Implications of a long-lived basal magma ocean in generating Earth's ancient magnetic field. This paper could serve as a starting point to discuss a potential dynamo in magma oceans of super-Earths. Another interesting paper in this context is that of Labrosse et al., *Nature* 450, 866 (2007), A crystallizing dense magma ocean at the base of the Earth's mantle, which discusses the influence of melting and of the mixing behavior in the deep Earth both on actuating and maintaining its geodynamo and its thermal history. From my point of view, both papers (there may be more) are relevant for the topic.

I really appreciate that the authors tried to perform HSE calculations for the conductivities as suggested which I consider as essential for a proper description of the physical processes in the

deep interior or magma ocean of rocky exoplanets. Unfortunately, this attempt has not been successful due to technical difficulties. The authors should continue this work. The arguments that the PBE results can still be used for this purpose are not valid. In addition, existing literature on the geodynamo generated in the magma ocean of the young Earth should be taken into account in this context. I conclude that the authors improved the paper in some points but could not address my major concern appropriately. According to the high standards of Nature Journals I recommend to reject the paper.

Reviewer #2 (Remarks to the Author):

After carefully reading the revised manuscript, added supplementary information as well as the answers to referees' questions and comments, I would like to suggest the paper: 'Electrical conductivity and magnetic dynamos in magma oceans of Super-Earths' to be published on Nature Communications.

We would like to thank both referees for considering the revised version of our manuscript. Since we could not completely satisfy Referee #1's request regarding the HSE conductivity calculations at that point, we decided to keep working on this issue. Since our last submission, we were able to remove the technical problems and finally we were successful in performing a complete conductivity calculation with the HSE functional for liquid MgO at 12000 K and 470 GPa. We hope this will fulfill referee #1's expectations.

Here are more detailed comments.

Reviewer #1 (Remarks to the Author):

The authors have submitted a revised version that takes into account most of my suggestions. My main concern was related to the results for the electrical and thermal conductivity of MgO, SiO₂, and MgSiO₃. The PBE XC functional used in the DFT-MD simulations underestimates the band gap systematically so that conductivities or optical properties predicted on that basis, in particular along a path from an insulator to a semiconductor or bad metal as relevant for these materials and planetary interior conditions, are disputable.

We agree that GGA functionals tend to underestimate the band gaps of insulators and that relying on them only is disputable. But, as we discussed in our previous reply, this is not the case for the present manuscript because, fortunately, for the materials under consideration, we are able to compare with laboratory measurements of the reflectivity. Our theoretical predictions for the reflectivities for MgO and MgSiO₃ are close to the experimental values – as can be seen in Fig. 4 of the manuscript –, which means that our conductivity calculations are reasonable as well. Moreover, the low pressure reflectivities are 0 which means that we actually retrieve the optical index of the uncompressed material in its insulating phase even with PBE. In our opinion, it is a proof that the PBE calculations are reasonable, or at least not incorrect. We see a deviation for SiO₂ but is understandable because there is a disagreement in the pressure.

The authors have followed my advise and evaluated the Kubo-Greenwood formula for a number of snapshots of the simulation using the HSE XC functional. The HSE calculations are, due the inclusion of parts of the non-local Hartree-Fock exchange, numerically much more expensive than the PBE calculations but predict the electronic band gap of many elements and compounds with much higher accuracy than the PBE XC functional can do.

However, the authors were only able to extract the DOS and not the conductivities due to technical difficulties. Based on this rather limited new information they claim that the HSE results essentially reproduce the previous PBE results. I do not agree with this conclusion.

We are now able to present a complete Kubo-Greenwood conductivity calculation using HSE06 for liquid MgO at 12000 K and 470 Gpa, please see Fig. 1 below. Since so far we had used Abinit for the conductivity calculations, we wanted to make sure that the results from both codes are in good agreement. As the referee can see, the agreement between VASP and Abinit is excellent except below ~1 eV but this is purely numerical and due to the way the calculation is performed in both codes. Now, as we anticipated, the HSE result is qualitatively similar as the PBE one. It shows a shifted maximum which corresponds to the shift in the highest peak in the valence states or said otherwise to the widening of the gap. The extrapolation at lower energy gives a lower DC conductivity. It actually goes down from 366 S/cm with PBE to 237 S/cm with HSE06. The decrease is thus of 36% while our estimation in our last reply based on the integrated DOS gave 32%. This simple prediction is thus very well corroborated.

Figure 1: Conductivity of liquid MgO at 12000 K and 470 GPa as a function of the excitation energy using different functionals.

We included this figure at the end of the Supplementary material and changed the HSE discussion. We also slightly modified the method section to stress our comparison with HSE.

The HSE DOS shows clearly a wider band gap than PBE, see Figs. 6 and 7 in

the SM. The occupation numbers and the dipole matrix elements entering the Kubo-Greenwood formula are influenced for those energies as well. Only a full HSE calculation would show the quantitative effect. The argument of the authors that the integrated DOS (this normalization gives the particle number or density) can serve as an estimate for the expected conductivity shifts does not apply in this case since only the conduction band is considered. This would give a good estimate for metallic states but not for semiconductors or semimetals as in the present case. A more detailed discussion on the conductivity should start from a proper transport theory and not from the DOS alone since disorder (via $S(k,\omega)$), screening (via $\epsilon(k,\omega)$) and collisions (ions, electrons, phonons) influence the conductivity. A simple Ziman theory where the DOS (or Fermi energy) plays the major role is not applicable for these conditions.

We have now performed the full HSE Kubo-Greenwood calculation that the referee requested above. We apologize for not presenting in our previous reply due to technical difficulties.

The authors repeatedly point out that the qualitative behavior, in particular the ratio of the conductivities for the solid and liquid phases, remains unchanged, regardless of using the PBE or the more suited HSE XC functional. It is clear that HSE would shift the conductivities of all materials for the interior conditions considered here to lower values with respect to PBE, both in the liquid and solid. The liquids would perhaps still have a higher dc conductivity than the solid phases. Better predictions for the transport properties would represent an important step forward in this context which, however, is not made in the paper.

We hope that our additional calculation of the conductivity will convince referee #1 that our results are robust and reliable for what they are. We do not claim the exact value of the conductivity is the PBE but that it is a very good estimate and that the trends we find on each silicate system we studied is correct and of significance. As we argued in the previous reply, even a decrease by 30% would not change the fact that a dynamo can happen in a magma ocean nor that it is likely to generate a multipolar field.

Since the conductivities (but also viscosities and the density and temperature profiles enter the dynamo simulations) are still not known accurately, performing a parameter study would be a reasonable alternative so that the values for which a dynamo in the magma ocean actuates could be isolated.

Interestingly, such a parameter study has already been done for the young Earth, see Ziegler and Stegman, *Geochem. Geophys. Geosyst.* 14, 4735 (2013), Implications of a long-lived basal magma ocean in generating Earth's ancient magnetic field. This paper could serve as a starting point to discuss a potential dynamo in magma oceans of super-Earths. Another interesting paper in this context is that of Labrosse et al., *Nature* 450, 866 (2007), A crystallizing dense magma ocean at the base of the Earth's mantle, which discusses the influence of melting and of the mixing behavior in the deep Earth both on actuating and maintaining its geodynamo and its thermal history. From my point of view, both papers (there may be more) are relevant for the topic.

The referee #1 points out two well-known references regarding the primitive Earth' magnetic field and the sources of convection in the Early Earth. There are other important references in the literature investigating these problems even for Super-Earths and we cited some of them. Because the number of citations is limited in Nature Communications, we decided not to cite these two because they are not directly investigating Super-Earths. They are also mostly investigating the source of convection and the properties of the flow while we wanted to investigate the minimum criteria for the convective activity to sustain geodynamo. We also do not argue that we solved the problem entirely. Our aim is to stimulate a new field of research in the area of magnetic fields in Super-Earths which have their own specificities.

I really appreciate that the authors tried to perform HSE calculations for the conductivities as suggested which I consider as essential for a proper description of the physical processes in the deep interior or magma ocean of rocky exoplanets. Unfortunately, this attempt has not been successful due to technical difficulties. The authors should continue this work. The arguments that the PBE results can still be used for this purpose are not valid. In addition, existing literature on the geodynamo generated in the magma ocean of the young Earth should be taken into account in this context. I conclude that the authors improved the paper in some points but could not address my major concern appropriately. According to the high standards of Nature Journals I recommend to reject the paper.

We hope that the HSE conductivity calculation will convince the referee of our results robustness and hope that he will reconsider his previous recommendation.

Reviewer #2 (Remarks to the Author):

After carefully reading the revised manuscript, added supplementary information as well as the answers to referees' questions and comments, I would like to suggest the paper: 'Electrical conductivity and magnetic dynamos in magma oceans of Super-Earths' to be published on Nature Communications.

We thank referee #2 for the positive evaluation.

REVIEWERS' COMMENTS:

Reviewer #2 (Remarks to the Author):

After carefully reading the legitimate concerns raised by the Reviewers (in particular Reviewer #1) and the answer proposed by the authors, I would consider the different issues well addressed by the authors and in the specific:

1) even with PBE-XC, the authors had already predicted in a reasonable range of error reflectivities data in comparison with experimental values. This has been also obtained for uncompressed materials where the PBE-XC tends to strongly underestimate the band gap;

2) the authors have filled the gap in their data by calculating by means of HSE06-DFT conductivities and comparing them with PBE results from two different ab-initio codes. I have also found quite useful that for such oxides it is possible to have a rough estimate (around 30%) simply by using PBE-XC;

3) More important, the same 30% overestimation does not affect in any way the main original conclusion of the paper about the dynamo in a magma ocean and the generation of a multipolar field

4) last note: it is well established that HSE06 (screened DFT) can predict with a very good approximation the band gap of semiconductor materials with respect to PBE-XC and free of experimental-wise parameters like DFT-U methods. In most cases the band gap itself calculated with hybrid-DFT methods is still an overestimation of the experimental data when compared with beyond DFT methods (GW). And accurate GW calculations are still computationally very expensive to be performed on supercell models. So, the comparison produced in this paper between PBE-XC and an affordable-state-of-the-art HSE06 can give us a reasonably narrow bracketing of the real conductivity behaviour.

In conclusion, for the novelty of the study, the wide range of techniques involved in the theoretical experiments, for the high quality of the results, I would suggest the manuscript: "Electrical conductivity and magnetic dynamos in magma oceans of Super-Earths" to be published on Nature Communications.

Reviewer #4 (Remarks to the Author):

The authors present conductivity calculations for magma materials made with the DFT-MD method. From the technical aspect this work seems to be reasonably well done, although I am not fully convinced that a high precision of the conductivities was achieved with the extrapolation of $\sigma(\omega)$ to zero as shown in the supplemental material. The influence of the exchange correlation functional, as requested by referee 1, was estimated.

However, I cannot recommend to publish this work in Nature communications because similar calculations for SiO₂ (Scipioni, PNAS 2017) and for FeMgO mixtures (Holmstrom, EPS Lett., 2018) are already published. These papers use the same method, find similar results, and come to similar conclusions as the authors of this paper. These papers are not cited in this manuscript. Thus, due to the lack of novelty, this work should clearly be published in a more specialized journal and comparison with the previous work needs to be made, too.

We would like to thank the two referees for considering the revised version of our manuscript. Below we addressed the arguments of the second referee and cite the two publications that were mentioned.

Reviewer #1 (Remarks to the Author):

After carefully reading the legitimate concerns raised by the Reviewers (in particular Reviewer #1) and the answer proposed by the authors, I would consider the different issues well addressed by the authors and in the specific:

- 1) even with PBE-XC, the authors had already predicted in a reasonable range of error reflectivities data in comparison with experimental values. This has been also obtained for uncompressed materials where the PBE-XC tends to strongly underestimate the band gap;
- 2) the authors have filled the gap in their data by calculating by means of HSE06-DFT conductivities and comparing them with PBE results from two different ab-initio codes. I have also found quite useful that for such oxides it is possible to have a rough estimate (around 30%) simply by using PBE-XC;
- 3) More important, the same 30% overestimation does not affect in any way the main original conclusion of the paper about the dynamo in a magma ocean and the generation of a multipolar field
- 4) last note: it is well established that HSE06 (screened DFT) can predict with a very good approximation the band gap of semiconductor materials with respect to PBE-XC and free of experimental-wise parameters like DFT-U methods. In most cases the band gap itself calculated with hybrid-DFT methods is still an overestimation of the experimental data when compared with beyond DFT methods (GW). And accurate GW calculations are still computationally very expensive to be performed on supercell models. So, the comparison produced in this paper between PBE-XC and an affordable-state-of-the-art HSE06 can give us a reasonably narrow bracketing of the real conductivity behaviour.

In conclusion, for the novelty of the study, the wide range of techniques involved in the theoretical experiments, for the high quality of the results, I would suggest the manuscript: "Electrical conductivity and magnetic dynamos in magma oceans of Super-Earths" to be published on Nature Communications.

We thank referee #2 for the positive evaluation.

Reviewer #4 (Remarks to the Author):

The authors present conductivity calculations for magma materials made with

the DFT-MD method. From the technical aspect this work seems to be reasonably well done, although I am not fully convinced that a high precision of the conductivities was achieved with the extrapolation of $\sigma(\omega)$ to zero as shown in the supplemental material. The influence of the exchange correlation functional, as requested by referee 1, was estimated.

However, I cannot recommend to publish this work in Nature communications because similar calculations for SiO₂ (Scipioni, PNAS 2017) and for FeMgO mixtures (Holmstrom, EPS Lett., 2018) are already published. These papers use the same method, find similar results, and come to similar conclusions as the authors of this paper. These papers are not cited in this manuscript. Thus, due to the lack of novelty, this work should clearly be published in a more specialized journal and comparison with the previous work needs to be made, too.

We thank the referee #4 for pointing out these two very recent publications. We cite them both in our revised manuscript. The reference to Scipioni et al. was added to our discussion of the SiO₂ conductivity. One should note that their values of the conductivities fully support our findings. They did not however specifically analyze the change in conductivity upon melting. This change for the same pressure-temperature conditions as well as the interpretation in terms disorder are central points in our manuscript. The discussion about the geodynamo is much more detailed in our article and should be of interest for many different communities and initiate further studies.

We also thank the referee for mentioning the Holmstrom et al. reference as it nicely supports one of the conclusions in our manuscript. The iron contents in magma ocean is one crucial parameter that needs to be addressed in future studies. We also need to point out differences between Holmstrom et al. and our work. We systematically study three different geomaterials and demonstrate that they all show very similar increases in conductivity upon melting. We propose there is a general trend among all geomaterials, which makes it relevant for all Super-Earth simulations. Arguments only based on (Mg,Fe)O alone are not as convincing. Compared to Holmstrom et al., we address not only the electric conductivity but also the thermal conductivity, the optical properties and the dynamo processes.

We also need to point out that Scipioni et al. and Holmstrom et al. articles were published only after we had submitted our manuscript for publication.